# Light induced non-volatile switching of superconductivity in single layer FeSe on SrTiO$_3$ substrate

Ming Yang[1,2], Chenhui Yan [1], Yanjun Ma[1], Lian Li[1] & Cheng Cen [1]

The capability of controlling superconductivity by light is highly desirable for active quantum device applications. Since superconductors rarely exhibit strong photoresponses, and optically sensitive materials are often not superconducting, efficient coupling between these two characters can be very challenging in a single material. Here we show that, in FeSe/SrTiO$_3$ heterostructures, the superconducting transition temperature in FeSe monolayer can be effectively raised by the interband photoexcitations in the SrTiO$_3$ substrate. Attributed to a light induced metastable polar distortion uniquely enabled by the FeSe/SrTiO$_3$ interface, this effect only requires a less than 50 $\mu$W cm$^{-2}$ continuous-wave light field. The fast optical generation of superconducting zero resistance state is non-volatile but can be rapidly reversed by applying voltage pulses to the back of SrTiO$_3$ substrate. The capability of switching FeSe repeatedly and reliably between normal and superconducting states demonstrate the great potential of making energy-efficient quantum optoelectronics at designed correlated interfaces.

[1] Department of Physics and Astronomy, West Virginia University, Morgantown, West Virginia 26506, USA. [2] National Key Laboratory of Science and Technology on Power Sources, Tianjin Institute of Power Sources, Tianjin 300384, P. R. China. These authors contributed equally: Ming Yang, Chenhui Yan. Correspondence and requests for materials should be addressed to L.L. (email: lian.li@mail.wvu.edu) or to C.C. (email: chcen@mail.wvu.edu)

Optical control of superconductivity has been the focus of many research works[1–5]. For instance, optical phonon pumping can be used to facilitate electron pairing[3–5]. Such method, however, usually requires stringent resonance conditions and intense laser pulses. The produced pairing states often have a short lifetime and are only detectable by ultrafast pump–probe measurements. To date, persistent impact on superconductivity generated by continuous-wave (CW) light has only been demonstrated in ionic liquid-gated organic materials[1], where the photochemically controlled isomerization of the ionic liquid modulated the doping level in the superconducting layer.

Here, we explore a different strategy to optically manipulate superconductivity in epitaxial heterostructures where the photo-active and superconducting layers are strongly coupled through the interface. By selecting a photoactive substrate that also exhibits strong electron correlations, photoexcitation can yield additional functionalities far beyond doping. The material system we study here is the heterostructure of $FeSe/SrTiO_3$. $SrTiO_3$, well known for its rich structural and electronic phases[6–9], has a 3.2 eV direct optical bandgap. In contrast, FeSe is a layer superconductor that can be isolated to a single layer[10]. Monolayer FeSe is particularly susceptible to charge transfer doping[11–13], strain[11,14], phonon scattering[15,16], and spin related proximity effects[17] at the interfaces. The combination of these two materials has led to $T_C$ well above the bulk values of FeSe[11,18,19]. In this system, we show that a higher-$T_C$ metastable state in FeSe can be reached by a weak CW UV photoexcitation in $SrTiO_3$ substrate, owing to a polaron related interface polar distortion. Using tailored sequence of UV irradiation and field-controlled dipole re-orientations, the heterostructure can be persistently driven between the metastable state and its ground state, allowing the superconducting zero-resistance to be rapidly turned on and off. This realization of persistent optical switching of high-temperature superconductivity in $FeSe/SrTiO_3$ highlights an interesting route toward the active manipulations of quantum materials.

## Results

### Superconducting transitions in FeSe persistently controlled by photoexcitations in $SrTiO_3$ substrates

One unit cell (uc) FeSe film (Fig. 1a) was grown on $TiO_2$-terminated insulating $SrTiO_3$ (001) substrate by molecular beam epitaxy (MBE). On top of the FeSe monolayer, ~10 uc FeTe capping layer was grown subsequently. Transport properties of the capped film were characterized shortly after the sample was removed from the ultrahigh

vacuum (UHV) growth chamber. A typical set of measurement data is shown in Fig. 1. In dark, the onset temperature ($T_C$) of the superconducting transition was around 24 K (Fig. 1b, black). Illuminated by 3.5 eV ultraviolet (UV) light, $T_C$ of the sample was raised to 30 K (Fig. 1b, red). The observed superconductivity enhancement can be produced by very weak UV light fields and is independent on the illumination intensity (Fig. 1c). Photoenergy-dependent measurements were also carried out at 1.5, 2.3, and 3.1 eV (Supplementary Figure 1). These longer wavelength lights produced no observable effect in the transport measurements. In metallic systems with large carrier densities, such as FeSe[20], the effects of direct carrier excitations by weak CW light on the electrical properties are usually negligible. Considering the stark contrast of the electrical responses to photons with energies below and above 3.2 eV (i.e., bandgap of $SrTiO_3$), the UV-induced superconductivity enhancement observed here more likely originated from the optical absorption in the $SrTiO_3$ substrate rather than in the gapless FeSe film.

The light-induced transition into superconducting state occurs very fast. While the study on the dynamic switching process is currently limited by the time needed to update the UV diode power supply output (a few ms), the almost instantaneous resistance drops observed upon turning on the UV light indicate that the light switching speed is at least in the kHz range or better. More interestingly, the light-induced superconductivity enhancement in FeSe persisted even after the UV light was turned off (Fig. 2a). As shown in Fig. 2b, following a sharp optical switching, the produced zero-resistance state was very stable in dark at 16 K. This state can remain at least for days when the sample temperature is kept constant. To better characterize the persistency of the light-induced superconductivity enhancement, we performed a multi-loop temperature variation experiment as illustrated in Fig. 2c. In this experiment, the zero-resistance state was first created at 16 K by UV light. After tuning off the UV light, the sample went through a series of thermal cycles to different maximum heating temperatures ($T_H$ = 20K, 30K, 40K, …, 300K). After reaching $T_H$, the sample was cooled back to 16 K where its resistance was measured. The resistance at 16 K as a function of $T_H$ exhibits a plateau-like behavior with two significant upturns at 40 K and 150 K, both coinciding with structural phase transitions in $SrTiO_3$. Around 40 K, $SrTiO_3$ undergoes a transition into quantum paraelectric state[6,21–24]. And at 150 K, a cubic–tetragonal structural phase transition occurs in the surface layers of $SrTiO_3$[25,26]. It's worth noting that the dwell

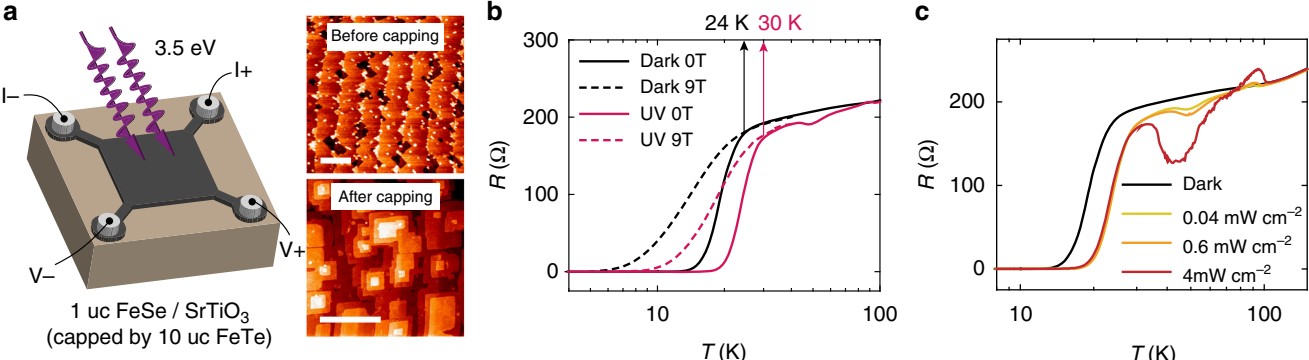

**Fig. 1** Raising $T_C$ by light in $FeSe/SrTiO_3$ heterostructure. **a** Effects of UV light on $FeSe/SrTiO_3$ were evaluated by transport measurements in van der Pauw geometry. Scanning tunneling microscope (STM) topography images of sample with and without FeTe capping layer are shown on the right. Scale bars indicate 100 nm. **b** Solid lines: sample resistances measured in dark (black) and under UV illumination (red), showing an increase in $T_C$ from 24 K to 30 K. Dashed lines plot the resistances measured with 9 T out-of-plane magnetic field. **c** Resistances measured at different UV intensities. Despite of the greater normal state resistance modulations at larger light intensities, the $T_C$ enhancement is intensity independent

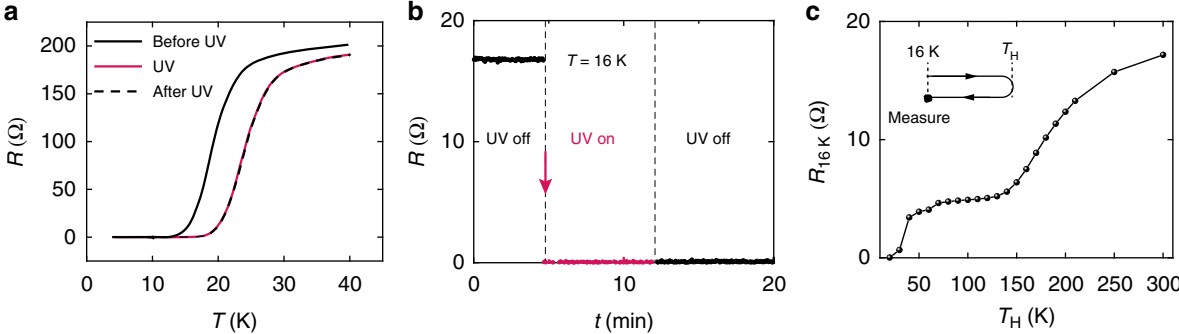

**Fig. 2** Persistent superconducting state generated by UV illumination. **a** Resistance measured before UV illumination (black, solid), during UV illumination (red, solid), and after turning off the UV light at base temperature (black, dashed). **b** Sample resistance measured at 16 K as the UV light was switched on and off. Resistance dropped to zero instantly when UV light was turned on and remained zero even after the light was turned off. **c** To study the retention of this effect, a series of thermal cycles of 16 K $\rightarrow T_H \rightarrow$ 16 K were performed after turning off the UV light. At the end of each thermal cycle, sample resistance at 16 K was measured and plotted as a function of the maximum temperature ($T_H$) reached during sample heating

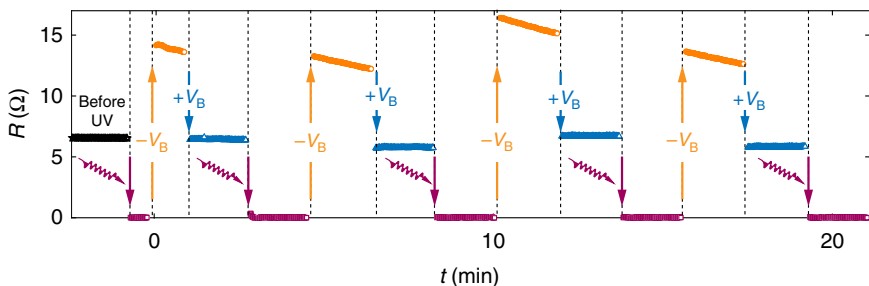

**Fig. 3** Switching between normal and superconducting states. Black stars plot the 17 K resistance measured before any UV exposure. Purple squares plot the zero resistance of a superconducting state achieved by a brief flash of UV light (with a duration less than 1 s). After applying five pulses of $V_B = -100$ V with 5 s duration to the back of the $SrTiO_3$ substrate, the normal state in FeSe can be recovered with a resistance (orange circles) that is larger than the pre-UV exposure value but slowly decreasing over time. The evolving resistance can be quickly stabilized near the pre-UV exposure value (green triangles) by applying one pulse of $V_B = 100$ V with 5 s duration. The non-volatile switching between normal and superconducting zero-resistance states by UV light and back bias pulses can be repeated many times without any sign of fatigue

time at $T_H$ had no observable effect on the measurement result. For example, after reaching 200 K, cooling immediately versus staying at 200 K for several hours before cooling yielded the same resistance measurement at 16 K.

Instead of thermal cycling to room temperature, the UV-induced transition to superconducting zero-resistance state can also be rapidly reverted at low temperature by applying bias pulses to the back of the 0.5-mm-thick $SrTiO_3$ substrate. To restore the finite resistance, a negative bias needs to be applied first. As shown in Fig. 3, after five −100 V, 5s voltage pulses, the sample resistance can be raised to a value even larger what was measured prior to UV exposure. This large resistance, however, was decreasing slowly over the time. Then, by applying a positive (5 s, 100 V) voltage pulse, the sample resistance can be quickly stabilized near the pre-exposure value. We note that, the seconds-level back bias switching speed, considerably slower than what was produced by UV light, might be related to the capacitive effects and defect states associated with the thick high-$k$ dielectric substrate. We also note that, the persistent switching effects of back biases are only possible after UV exposure. In as-grown samples, the effects of back biases are completely volatile as discussed in ref. [13]. Using the combination of UV light and back bias pulses, FeSe can be switched between superconducting and normal states repeatedly and reliably.

**Photoexcited carriers transfer between $SrTiO_3$ and FeSe.** The mechanism for the observed optical control of superconductivity can be complex, as UV light affects $SrTiO_3$ in many ways. For

example, if defects are introduced to $SrTiO_3$ by the substrate pre-annealing step in UHV, the relaxation of defects after carrier excitation can serve as hole traps and produce persistent UV photoconductance[27,28]. Exposing freshly cleaved $SrTiO_3$ (001) surface to intense extreme ultraviolet (EUV) light can lead to the formation of surface two-dimensional electron gas (2DEG)[29–31], possibly originating from the creation of either oxygen vacancies or surface reconstructions. Additionally, due to the strong coupling between photocarriers and the lattice in $SrTiO_3$, UV irradiations were also found to excite soft phonons/polarons[32,33] and cause persistent phonon softening at low temperatures[34]. In FeSe/$SrTiO_3$ heterostructures, many of these effects can potentially impact the superconducting behaviors in FeSe through the interface, such as generating charge transfer doping[11–13], modulating interface electron–phonon coupling[15,16], and producing interface lattice distortions[14,35–39]. Among these possible contributors to the observed UV-induced persistent superconductivity enhancement, we first evaluate the role of photo-excited charge transfer.

As shown in Fig. 4c, without UV illumination (black curve), the magnitude of the Hall resistance $R_H$ decreases sharply toward zero below $T_C$ due to the Meissner effect. In the presence of light (yellow and red curves), the enhancement of superconductivity is evident by the higher temperature where this sharp drop occurs. Above $T_C$, the Hall resistance under illumination only deviates from the dark value between 50 K and 90 K. Such changes, however, did not persist after turning off the UV light. In this temperature range, the field dependence of the transverse

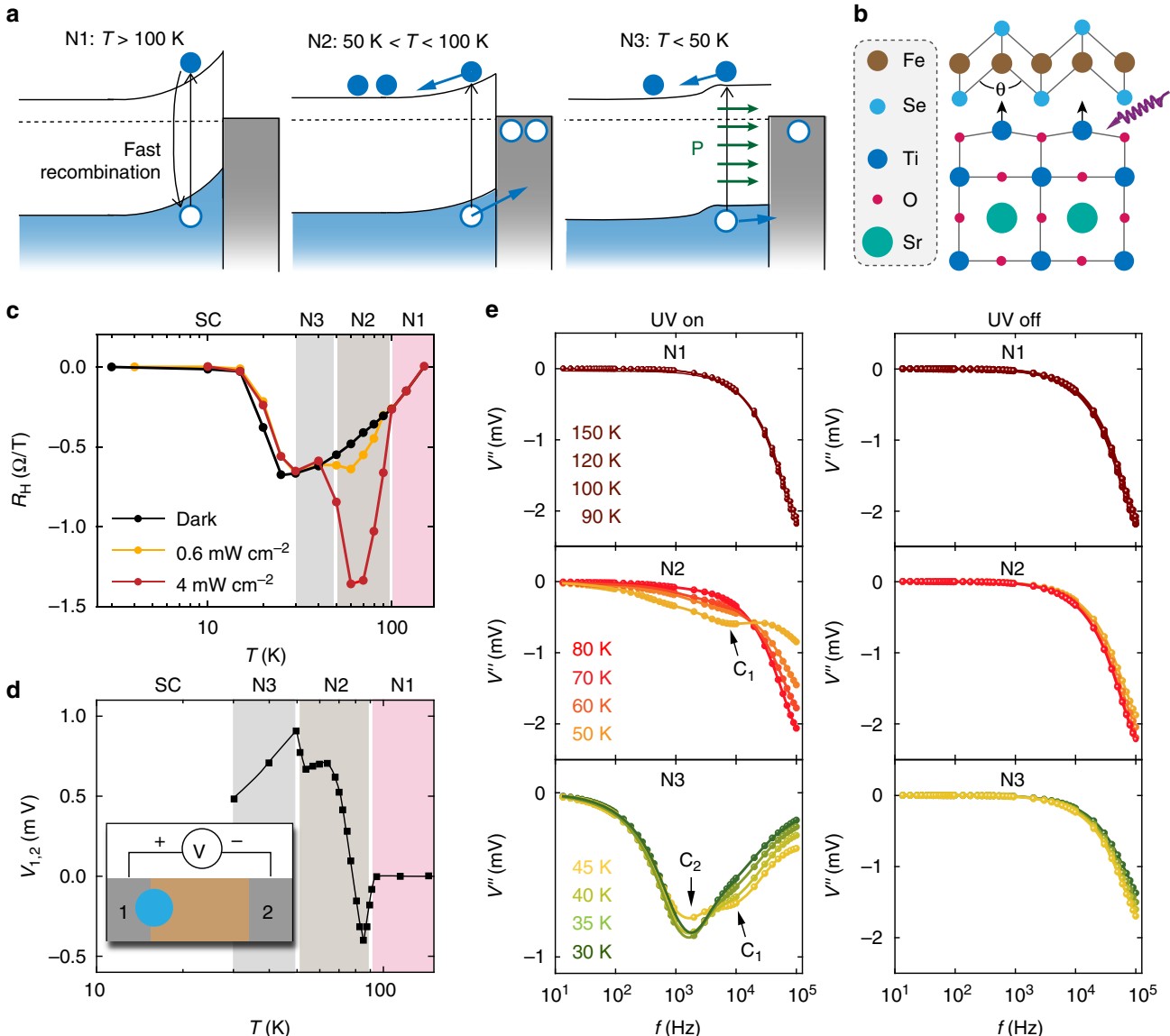

**Fig. 4** Photoinduced charge transfer and superconductivity related interface polar distortion. **a** Schematics of the band bending and UV excited charge transfer in the FeSe/SrTiO$_3$ heterostructure at different temperature ranges. **b** Instead of photodoping, UV-induced SrTiO$_3$ surface polar distortion is more likely the cause of the enhanced superconducting behaviors. A reported double-TiO$_2$ layer terminated SrTiO$_3$ surface structure[13, 39] is depicted. **c** Hall resistances measured as functions of temperature with and without UV illumination. **d** Temperature dependence of open circuit photovoltage measured between two surface electrodes when the UV light was focused near one of them. **e** Frequency spectra of the out-of-phase impedance responses at various temperatures during and after the UV exposure. R-C resonances marked by C$_1$ and C$_2$ indicate the emergences of photocapacitance at low temperatures

resistance became nonlinear under the UV exposure (Supplementary Figure 3), indicating the formation of mobile electrons with mobilities much higher than the intrinsic carriers in FeSe. This effect was accompanied by a large light-induced magnetoresistance (Supplementary Figure 3) and a surface UV photovoltage that becomes more positive with decreasing temperature (Fig. 4d).

These features between 50 K and 90 K can be understood by a photoexcited charge transfer at the FeSe/SrTiO$_3$ interface. When electron–hole pairs are generated in SrTiO$_3$ by UV light, due to its upward band bending toward the interface[12,13], holes are driven into FeSe, leaving behind electrons in SrTiO$_3$ (Fig. 4a, middle). Such spatial charge separation effectively prevents the electron–hole recombination, giving rise to a positive surface photovoltage and a unique n-type photoconductance in SrTiO$_3$ that was not observed in identically processed bare SrTiO$_3$ substrates within this temperature range (Supplementary Figure 4).

While these results clearly indicate a significant photocarrier transfer at the FeSe/SrTiO$_3$ interface, the resultant hole doping into FeSe is not likely to enhance superconductivity. Several studies have identified the inter-electron-pocket pairing as the most likely mechanism for the high-$T_C$ state in FeSe[40–42], which is expected to be enhanced by electron rather than hole doping[11,12,43]. Also, light-induced Hall resistance modulation (Fig. 4c) and photovoltage (Fig. 4d) both decreased significantly below 50 K, indicating that the UV-induced interface charge transfer is suppressed in the temperature range that is most relevant to the superconducting transition.

**Photocapacitances and photocarriers mediated interface distortions.** To identify UV-induced effects in SrTiO$_3$ at temperatures below 50 K that can positively impact the superconducting state in FeSe, AC impedance spectroscopy was performed

(Fig. 4e) to detect signals that may not manifest in DC ohmic measurements. Without UV exposure, the sample exhibits an ~3 nF temperature-independent capacitance (Supplementary Figure 5) that is related to the geometric dielectric properties of the sample and contact junctions. Above 100 K, sample capacitance was not affected by the UV light (Fig. 4e, top). Between 50 K and 90 K, a $10^1$ nF level photocapacitance (marked as $C_1$ in Fig. 4a) was measured and associated with the interface charge separation (Fig. 4e, middle). Below 50 K, while the charge transfer is diminished significantly, another much greater (~$10^2$ nF) photocapacitance (marked as $C_2$ in Fig. 4a) emerges (Fig. 4e, bottom) and remains strong as temperature falls even below $T_C$ (Supplementary Figure 5b).

The observed large photocapacitance provides a valuable hint for understanding the optical superconductivity enhancement. The generation of capacitance relies on the changes of either free or bound charge distributions. Since the free charge separation between different material layers are significantly suppressed at low temperatures, the onset of photocapacitance $C_2$ mostly likely originates from an enhanced material polarizability and the related bound charge formation in response to UV light. Consistent with such assessment, a giant enhancement in dielectric constant was reported in $SrTiO_3$ samples irradiated by UV light at low temperatures[44]. As discussed above, $SrTiO_3$ undergoes a quantum paraelectric phase transition at low temperatures[6,21–24], where quantum fluctuations associated with zero-point energy prevent the onset of long-range ferroelectric order. In this phase, photoexcited electrons can quadratically couple to the $T_{1u}$ soft mode (relative displacement between the Ti ion and the oxygen octahedra) and directly impact the quantum fluctuations[32–34,45]. In particular, the polarons formed from photocarriers and phonons can serve as effective charge trap to suppress electron–hole recombination and generate local dipole moments[44], leading to the large photocapacitance observed in our experiments.

In $FeSe/SrTiO_3$ heterostructure, the interface band bending[12,13] gives rise to a large electric field at the interface that points from $SrTiO_3$ to FeSe. In the presence of this field, the alignment of induced dipole moments at the interface may effectively modify the ferroelectric distortions with relative out-of-plane shifts between the Ti and O ions that are well known for both single-[46] and double-$TiO_2$[47] terminated $SrTiO_3$ surfaces. As a result, it is viable that the Se–Fe–Se angle in the FeSe monolayer, a parameter sensitively modulating the electron correlation strength in FeSe[35], will be perturbed as well (Fig. 4b). Because of such photocarrier-mediated structural distortion, an enhancement in superconductivity can be produced. Besides of producing a superconductivity enhancement in FeSe and a large photocapacitance in $SrTiO_3$, the $T_{1u}$ polaron related interface polarization will also reduce the band bending and thus suppress the photocarrier transfer between $SrTiO_3$ and FeSe (Fig. 4a, right). This effect well explains the reductions of surface photovoltage and Hall resistance modulations observed below 50 K (Fig. 4c, d).

We note that, the possibility of generating oxygen vacancies (OV) in $SrTO_3$ by UV irradiation was also suggested in several studies[30,48]. While OVs can have a lower formation energy[38] in the double-$TiO_2$ surface layer found in some $FeSe/SrTiO_3$ heterostructures[13,39] and are often associated with interface-enhanced superconductivity[37,49], they are not likely the origin of the light-induced $T_C$ enhancement observed here. First, the creation of OVs typically requires extended exposure time and should be highly dependent on light intensity[30,48], both of which are inconsistent with our observations. In addition, since FeSe is prone to oxidation, the removal of oxygen with FeSe in proximity may generate chemical modifications in FeSe that are harmful to its superconductivity.

As shown in Fig. 2, the light-induced zero-resistance state well persisted after the removal of UV light, indicating that the charge trapping by polarons and the interface polar distortion triggered by light is metastable in the quantum paraelectric phase without external perturbation. While the combination of photocarrier, strong electron–phonon coupling, and built-in electric field allows the system to reach such metastable state with higher $T_C$, there can be multiple pathways for the system to go back to the ground state. One of them is through thermal cycling to different structural phases (Fig. 2c). Alternatively, by applying an external field antiparallel with the built-in field, as executed via negative back biases (Fig. 3), re-alignment of the interface dipoles can also quickly excite the system from the UV-induced metastable state, allowing it to either slowly relax toward the ground state or rapidly settle with the aid of positive back bias (Fig. 3).

**Effects of the capping layer in restricting the superconducting transition temperature.** $T_C$ observed by ex situ transport measurements typically varied between samples when they were measured in as-grown state in dark (Fig. 5a, black). Nonetheless, the $T_C$ enhancement induced by UV light always saturated at around 30 K (Fig. 5a, red). To understand the origin of such $T_C$ saturation, we characterized the superconducting transition in the same sample both before it was capped by FeTe layers using in situ ARPES (Fig. 5b–d) and after it was capped using ex situ transport measurements (Fig. 5a, Supplementary Figure 6). Comparing with the ~50 K gap opening temperature detected by ARPES before capping, the ~30 K onset $T_C$ observed in ex situ transport measurements was much lower. Additionally, ARPES data showed that the uncapped film had no hole pocket at the Γ-point, and the n-type doping was little changed as temperature varied (Supplementary Figure 6a, b). In contrast, the ex situ transport measurements performed on capped film revealed a co-existence of electrons and holes and a strongly temperature-dependent doping type (Supplementary Figure 6c).

Since the FeTe capping layer itself typically had a much lower conductance comparing with the FeSe samples (Supplementary Figure 6d), electrical shunting caused by the capping layer is expected to be weak. Instead, the capping layer as well as the environmental factors more likely introduced a profound modification to the electrical properties of the FeSe monolayer underneath, which restricted the highest $T_C$ that can be reached by manipulating the $FeSe/SrTiO_3$ interface alone. To evaluate the full capability of the UV-induced superconductivity enhancement, alternative capping methods (Supplementary Figure 8) and/or in situ measurements with light excitations needs to be explored in the future.

In conclusion, we have shown that a brief exposure to a weak (~$10^1$ μW cm$^{-2}$) 3.5 eV CW UV light can raise the superconducting $T_C$ in $FeSe/SrTiO_3$ heterostructure and generate zero-resistance state that persists in dark for at least days. We attribute this effect to the strong photocarrier–phonon coupling in $SrTiO_3$ and the resultant metastable polar lattice distortion occurred at the $FeSe/SrTiO_3$ interface. Quick and non-volatile switching between this metastable state and the as-grown ground state can be achieved by UV photoexcitation and field-controlled interface dipole re-orientations. An ex situ $T_C$ upper limit of ~30 K, even with the UV enhancement, was observed. This $T_C$ bottleneck is a result of the second interface formed between FeSe and the capping layer. To further enhance the ambient superconducting performance of FeSe, this interface deserves as much as attention as the interface formed with the substrate. The $FeSe/SrTiO_3$ heterostructures also exhibited very large photovoltage and photocapacitance that are consequences of interfacial effects,

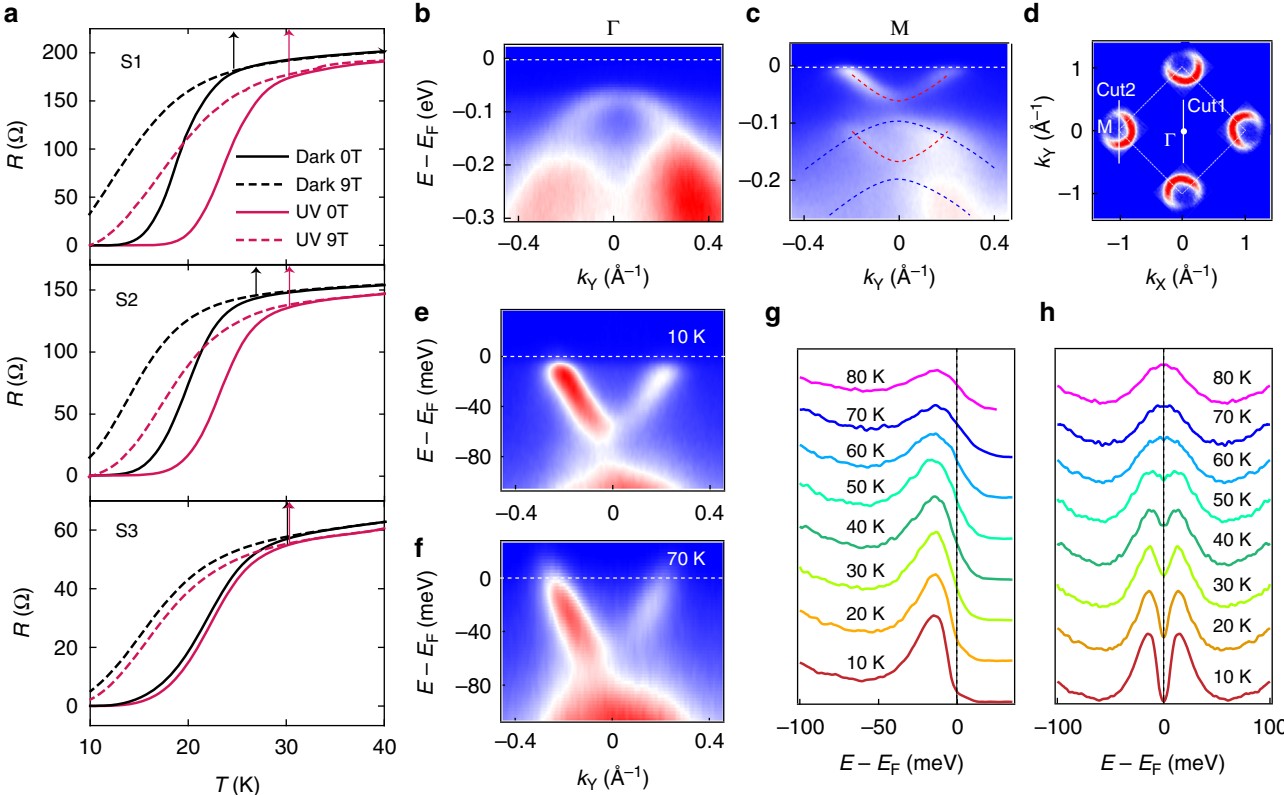

**Fig. 5** Capping layer related $T_C$ bottleneck. **a** Ex situ resistance of three capped samples (S1, S2, S3) measured with and without UV exposure. While the ex situ $T_C$ measured in darkness varied between different samples, the UV enhancement of $T_C$ always saturated at around 30 K, which was still much lower than what was measured in uncapped films, indicating the leading bottleneck of $T_C$ in capped samples is likely related to the capping layer. **b**, **c** In situ ARPES spectra of uncapped film (sample S3 before capping) taken across Γ-point (**b**) and M-point (**c**) along the directions as marked in the 2D intensity plot (**d**). **e**, **f** Spectra across M-point measured at 10K (**e**) and 70K (**f**). **g**, **h** Temperature-dependent energy distribution curves (EDCs) (**g**) and symmetrized EDCs (**h**) at the Fermi crossing, showing the superconducting gap opening between 50 K and 60 K

showing a promising potential for the implementations of designed heterostructures from 3D oxides and 2D layered materials in optoelectronic applications. Most importantly, the capability of manipulating the superconducting state by photo-excitations in the substrate demonstrated a promising venue for realizing active controls in correlated materials.

## Methods

**SrTiO₃ substrate preparation.** The insulating single crystal SrTiO₃ (001) was cleaned with deionized water for 30 min then chemical etched with buffered-oxide etchant for 1 min. It was subsequently thermal annealed in a tube furnace under $O_2$ flow for 4 h to obtain a TiO₂ terminated surface. The substrate was then transferred into the ultrahigh vacuum (UHV) MBE chamber and annealed at 600 °C for 30 min. After those treatment, the SrTiO₃ (001) surface becomes atomically flat with well-defined step-terrace structure, as shown in Supplementary Figure 2a.

**FeSe films and FeTe capping layer preparation.** The growth of monolayer FeSe films were carried out on SrTiO₃ (001) substrates in an UHV system (base pressure ~1 × 10⁻¹⁰ Torr) that integrates two MBE chambers, a room temperature scanning tunneling microscope (STM), a low temperature (5 K–300 K) STM and an angle revolved photoemission spectroscopy (ARPES). For the growth of monolayer FeSe films, the substrate was held at around 400 °C, and Fe and Se were supplied via separate Knudsen cells with a flux ratio of around 1:10. The film, as shown in Supplementary Figure 2b, was then annealed at ~520 °C for 2–3 h and in situ transferred into RT-STM and ARPES. For ex situ electrical transport measurements, around 10 ML FeTe films were deposited on 1 ML FeSe/SrTiO₃(001) by co-evaporating Fe and Te with a flux ratio of around 1:4, and the substrate was kept at ~300 °C. The growth rate is 0.5 ML/min for both FeSe and FeTe growth.

**STM and ARPES measurements.** In situ STM imaging with a chemically etched tungsten tip was used to monitor surface morphology of SrTiO₃ (001), FeSe and FeTe films at room temperature. ARPES was carried out with a Scienta DA30 analyzer and He discharge lamb ($hv = 21.218$ eV). The energy resolution was set at

~ meV, as shown in Supplementary Figure 6. The angular resolution is 0.3°. The Fermi level was determined by measuring the Ag film on Si (111) substrate (shown in Supplementary Figure 7).

**Magnetotransport and photocapacitance measurements.** Ex situ magneto-transport measurements with and without UV illumination was performed using a Quantum Design Physical Property Measurement System (PPMS). Photo-capacitance measurements were performed in the same PPMS system but using an external lock-in amplifier for the AC current sourcing and voltage detection. Indium electrical contacts were mechanically attached to the sample surface in Van der Pauw geometry at room temperature in the atmosphere. 3.5 eV UV light used in the experiments was generated by a light emitting diode (LED) and collimated by lenses.

**Photovoltage measurements.** Photovoltage measurements were performed in a Montana instrument Cryostation system. 3.5 eV UV light used was generated by a LED source and focused to the sample surface near one electrode by lenses. Voltage between illuminated and unilluminated electrodes was measured by a Keithley DC nanovoltmeter.

## Data availability

The datasets generated during and/or analyzed during the current study are available from the corresponding authors on reasonable request.

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

## Acknowledgements

Work performed by C.C.'s group is supported by Department of Energy grant no. DE-SC-0010399 and National Science Foundation grant no. NSF-1454950. Work performed by L.L.'s group is supported by Department of Energy grant no. DE-SC0017632.

## Author contributions

M.Y. and C.C. performed the magnetotransport, photovoltage, and photocapacitance measurements. C.Y. and L.L. performed the MBE growth, STM measurements, and ARPES measurements. Y.M. performed part of the substrate treatment. C.C. and L.L. conceived and organized the study. M.Y. and C.Y. analyzed the data and wrote the paper. All authors discussed the results and commented on the paper.

## Additional information

**Competing interests:** The authors declare no competing interests.

