## [Peer Review File · Nature Communications]

Reviewers' comments:

Reviewer #1 (Remarks to the Author):

Ming Yang et al. report a metastable state of enhanced superconductivity at the FeTe/FeSe/STO heterostructure. At specific temperature range, ultra-violet light illumination turns the sample into zero resistive state, and applying voltage pulses at the back gate switches it back to resistive state. By thermal cycling, Hall resistance, photovoltage, and AC impedance measurements, the authors conclude that the higher T_c is from interface polar distortion triggered by light.

Overall, the manuscript is well written, and the finding of light controlled superconductivity is quite intriguing and potentially useful in quantum optoelectronic applications. Therefore, I would recommend its publication in "Nature Communications" if the authors could address the following points:

1. The authors claim that photon energy lower than 3.2eV does not enhance the T_c . This is an important point in the story, and the authors should show data to support this claim.
2. As the T_c enhancement is not dependent on the illumination intensity, does the excitation photons in ARPES measurement switch the sample into "high T_c " state?
3. The T_c of monolayer FeSe/STO in this manuscript (Dark OT) is lower than that reported in literature (Chinese Physics Letters 31, 017401), which also use the FeTe to cap single-layer FeSe/STO. The measured T_c here is actually close to FeTe_{0.5}Se_{0.5}, which could show the highest T_c up to 21K (Physica C 514 (2015) 423–434). The author should exclude the possibility that the measured superconductivity is from the intermixing of FeTe/FeSe but not from the FeSe/STO interface. Besides, before claiming the T_c enhancement is from the optical absorption in STO, the authors need to discuss whether FeTe/FeSe interface or its intermixing could have a response to the UV light.

Reviewer #2 (Remarks to the Author):

This paper reports a new type of light-induced switching of superconductivity in single layer FeSe. The superconductivity of FeSe on SrTiO₃ was able to be turned on by UV light pulse, and to be turned off by gate bias treatments. This is highly original work and all the data presented here are of high quality and well organized. The authors also speculate that the microscopic mechanism of switching is based on lattice deformation of the FeSe layer, which seems reasonable and can be justified by capacitance, Hall effect, asymmetric UV excitation, and ARPES measurements.

I would suggest authors to add any data on the switching speed, as readers will be very much interested in such an information. Other parts of the paper are perfectly organized and documented. So, I strongly recommend the publication of this paper in Nature communications after minor revision suggested above.

Reviewer #3 (Remarks to the Author):

The authors present an interesting study of UV-enhanced superconducting transition in FeSe/STO heterostructure that is also metastable and non-volatile. The mechanism behind this novel phenomenon is attributed to a polar distortion of the interface induced by the polarons formed between photo-excited electrons and phonons. The proposed mechanism is consistent with various transport measurement results, and in line with the previous observation of giant enhancement of dielectric constant in STO by UV irradiation.

Overall, I don't find any fault with the experimental data and the discussions. The only comments I

have are about the abstract and the introduction. Certain phrases in these sections read weird, such as:

"Since effective superconducting pairing and great photoexcitation efficiency rarely coexist in a single material, this task is usually very challenging"

"ultrafast transient measurements"

"sustained superconductivity control"

"Choosing a photoactive layer that also has strong electron correlations, the functionalities of photoexcitations can be greatly extended beyond doping control"

"new venues toward active quantum logics".

If the authors could polish the abstract and the introduction, I think the manuscript will read a lot better.

Reviewer #1 (Remarks to the Author):

Ming Yang et al. report a metastable state of enhanced superconductivity at the FeTe/FeSe/STO heterostructure. At specific temperature range, ultra-violet light illumination turns the sample into zero resistive state, and applying voltage pulses at the back gate switches it back to resistive state. By thermal cycling, Hall resistance, photovoltage, and AC impedance measurements, the authors conclude that the higher T_c is from interface polar distortion triggered by light.

Overall, the manuscript is well written, and the finding of light-controlled superconductivity is quite intriguing and potentially useful in quantum optoelectronic applications. Therefore, I would recommend its publication in “Nature Communications” if the authors could address the following points:

Response: We thank the referee for the positive comments. Below we list our detailed responses. The changes we made based on your suggestions have allowed us to greatly improve the quality of the manuscript.

1. The authors claim that photon energy lower than 3.2eV does not enhance the T_c . This is an important point in the story, and the authors should show data to support this claim.

Response: We have included one figure (Fig. S1) in the supplementary materials to show the electrical responses of a 1 μ c FeSe/SrTiO₃ sample capped by 10 μ c FeTe to lights with photon energies from 1.5 eV to 3.1 eV, with the maximum phonon energy very close to but still falls below the band gap of SrTiO₃ (3.2 eV). As can be seen from the data, no obvious resistance change was detected for these visible laser excitations, supporting the attribution of the observed T_c modulation induced by 3.5 eV UV light to the inter-band photoexcitation in SrTiO₃.

2. As the T_c enhancement is not dependent on the illumination intensity, does the excitation photons in ARPES measurement switch the sample into “high T_c ” state?

Response: The reviewer is correct to point out that the EUV light source used in ARPES measurement also has enough energy to produce inter-band photoexcitations in SrTiO₃. However, at this point, we cannot conclusively determine the effects of ARPES light on the superconducting state of FeSe for two technical reasons. First, due to the limitation of our instrument, we cannot directly compare the in-situ transport properties with and without ARPES light excitations. Second, when the sample measured by ARPES was transferred to our PPMS system, it would have already been thermal-cycled to room temperature, which, according to our data as shown in Figure 2, is expected to wipe out the light induced persistent effect at low temperatures. Despite of these technical limitations, the reviewer does raise an important question of whether the EUV light excitations may cause differences in T_c measurements obtained by ARPES and other experiments performed in dark, such as STM or in-situ transport. We are currently upgrading our experimental set up to hopefully provide more direct experimental evidences in the future. In the meanwhile, we also would

like to point out that, the photoexcitations induced by an EUV light can be much more complicated than what's produced by the 3.5 eV UV light used in our current experiments. The EUV photons with much higher energies may give rise to a very different carrier energy distribution and may also lead to the possible generation of defects. These effects beyond simple inter-band photoexcitation will also need to be considered in future experiments.

3. The T_c of monolayer FeSe/STO in this manuscript (Dark 0T) is lower than that reported in literature (Chinese Physics Letters 31, 017401), which also use the FeTe to cap single-layer FeSe/STO. The measured T_c here is actually close to FeTe_{0.5}Se_{0.5}, which could show the highest T_c up to 21K (Physica C 514 (2015) 423–434). The author should exclude the possibility that the measured superconductivity is from the intermixing of FeTe/FeSe but not from the FeSe/STO interface. Besides, before claiming the T_c enhancement is from the optical absorption in STO, the authors need to discuss whether FeTe/FeSe interface or its intermixing could have a response to the UV light.

Response: We agree with the referee that the intermixing of Te with Se can occur during the growth of FeTe capping layers under Te-rich condition, which, for example, was reported by STEM studies (PRB 91, 220503(R) (2015)). However, we don't think such intermixing is the leading contributor to the superconductivity observed in our samples for the following reasons:

First, in our study, we have clearly and consistently observed T_c above 50 K by ARPES in samples prior to any FeTe capping, proving the existence of superconductivity in absence of Se-Te intermixing.

Second, while the ex-situ T_c measured in samples with monolayer FeSe and 10 uc FeTe capping layer are relatively lower than the record literature values, we were able to observe a much-enhanced T_c in samples with thicker capping layers. We have added such data to the supplementary materials (Figure S8). These samples, although subject to the same Se-Te intermixing that can occur at the FeSe/FeTe interface, exhibited T_c considerably higher than what was reported for FeTe_{0.5}Se_{0.5} system. Such measurements indicate that the FeSe/FeTe interface and the anion intermixing is not likely the main contributor to the superconductivity observed in our samples, whereas other effects such as better shielding of the FeSe/SrTiO₃ interface from the ambient environment and the second interface formed with FeTe, the quality of the SrTiO₃ surface preparation, FeSe growth conditions, and capping layer growth conditions may have played more important roles in determining the final ex-situ values of superconducting T_c .

Third, unlike the superconducting behaviors observed in bulk or thick films of FeTe_{1-x}Se_x, the situation of a-few-layer films can be very different. According to the STS studies reported in the same paper mentioned above (PRB 91, 220503(R) (2015)), while monolayer layer FeTe_{1-x}Se_x film grown on SrTiO₃ is also superconducting, bilayer or few-layer FeTe_{1-x}Se_x films are not superconducting at all, indicating the more critical role played by the FeTe_{1-x}Se_x/SrTiO₃ interface comparing to Se-Te intermixing in these ultrathin films.

Regarding the photoresponse from the FeSe/FeTe interface, while such effect might be present, it cannot be what's responsible for the T_c modulation observed. For metallic and narrow-bandgap systems such as FeSe and FeTe (as well as their interfaces), the carrier excitations can be produced by longer wavelength light much more efficiently comparing to short wavelength light, which is inconsistent with the photoenergy threshold required to impact the superconducting state as observed in our experiments (Fig. S1). Also, in systems with large carrier densities, such as FeTe/FeSe, the effects of direct carrier excitations by weak CW light on the electrical properties are usually negligible. Therefore, the origin of light induced T_c enhancement is more likely from the optical adsorption in SrTiO₃ rather than the FeTe/FeSe interface.

Reviewer #2 (Remarks to the Author):

This paper reports a new type of light-induced switching of superconductivity in single layer FeSe. The superconductivity of FeSe on SrTiO₃ was able to be turned on by UV light pulse, and to be turned off by gate bias treatments. This is highly original work and all the data presented here are of high quality and well organized. The authors also speculate that the microscopic mechanism of switching is based on lattice deformation of the FeSe layer, which seems reasonable and can be justified by capacitance, Hall effect, asymmetric UV excitation, and ARPES measurements.

I would suggest authors to add any data on the switching speed, as readers will be very much interested in such an information. Other parts of the paper are perfectly organized and documented. So, I strongly recommend the publication of this paper in Nature communications after minor revision suggested above.

Response: We thank the referee for the valuable comments and the recommendation for publication. Two sentences have been added to the manuscript to address the switching speeds associated with the UV light and electrical back biases, which are copied below:

“The light induced transition into superconducting state occurs very fast. While the study on the dynamic switching process is currently limited by the time needed to update the UV diode power supply output (a few ms), the almost instantaneous resistance drops observed upon turning on the UV light indicate that the light switching speed is at least in the kHz range or better.”

“We note that, the seconds-level back bias switching speed, considerably slower than what was produced by UV light, might be related to the capacitive effects and defect states associated with the thick high-k dielectric substrate.”

Reviewer #3 (Remarks to the Author):

The authors present an interesting study of UV-enhanced superconducting transition in FeSe/STO heterostructure that is also metastable and non-volatile. The mechanism behind this novel phenomenon is attributed to a polar distortion of the interface induced by the polarons formed between photo-excited electrons and phonons. The proposed mechanism is consistent with various transport measurement results, and in line with the previous observation of giant enhancement of dielectric constant in STO by UV irradiation.

Overall, I don't find any fault with the experimental data and the discussions. The only comments I have are about the abstract and the introduction. Certain phrases in these sections read weird, such as:

"Since effective superconducting pairing and great photoexcitation efficiency rarely coexist in a single material, this task is usually very challenging"

"ultrafast transient measurements"

"sustained superconductivity control"

"Choosing a photoactive layer that also has strong electron correlations, the functionalities of photoexcitations can be greatly extended beyond doping control"

"new venues toward active quantum logics".

If the authors could polish the abstract and the introduction, I think the manuscript will read a lot better.

Response: We thank the referee for the valuable suggestions. The following changes have been made in the abstract and introduction to improve the clarity of the discussion:

1. "Since effective superconducting pairing and great photoexcitation efficiency rarely coexist in a single material, this task is usually very challenging" has been changed into "Since superconducting materials rarely exhibit strong photoresponses, and vice versa, optically sensitive materials are often not superconducting, the efficient coupling of these two characters in a single material can be a challenging task."
2. "ultrafast transient measurements" has been changed to "ultrafast pump-probe measurements".
3. "sustained superconductivity control" has been changed to "persistent impact on superconductivity".
4. "Choosing a photoactive layer that also has strong electron correlations, the functionalities of photoexcitations can be greatly extended beyond doping control" has been changed to "By selecting a photoactive substrate that also exhibits strong electron correlations, photoexcitation can yield additional functionalities far beyond doping."

5. “new venues toward active quantum logics” has been changed to “new routes toward the active manipulations of quantum materials”.

REVIEWERS' COMMENTS:

Reviewer #1 (Remarks to the Author):

The authors have replied in a satisfactory way to my questions and concerns. They have revised the manuscript following my suggestions. I recommend the publication in Nature Communications in the current form.

REVIEWERS' COMMENTS:

Reviewer #1 (Remarks to the Author):

The authors have replied in a satisfactory way to my questions and concerns. They have revised the manuscript following my suggestions. I recommend the publication in Nature Communications in the current form.

We thank the reviewer for the time and the publication recommendation.